# Industrial technological advance and the employment demand for China's labour force —Micro evidence from labour hiring in companies

Jianmin Liu[1], Yifeng Shen[1]*, Wenye Fan[1], Xiya Wu[2]

1 School of Economics & Management, Nanchang University, Nanchang City, Jiangxi Province, P.R. China,
2 School of Economics & Management, East China Jiaotong University, Nanchang City, Jiangxi Province, P.R. China

* 15855647956@163.com

**Data Availability Statement:** All relevant data are within the manuscript and its Supporting.

## Abstract

Technological advance in industry has complexity, divisibility, systematicity and market selectivity, and companies may generate technological investment expansion and form technological job demand, which may also lead to the occupation of company resources and trigger the replacement of the original jobs. This paper empirically examines the impact of technological advance in industry on the labour employment demand of companies by taking Chinese A-share listed companies as research samples from 2008 to 2022, and finds that: overall technological advance in industry has a suppressive effect on the labour employment demand of companies; The heterogeneity test shows that industrial technological advancement mainly produces inhibitory effects on the labour employment of low-education level employees and production sectors, while it produces incentive effects on the labour employment of high-education level employees and non-production sectors Industrial technological advancement mainly produces job substitution and destructive effects on the labour employment of low-education level employees and production sectors, while it produces incentive effects on the labour employment of high-education level employees and non-production sectors. Mechanism test shows that industrial technological advancement has incentive effects such as technology investment expansion effect and industry chain conduction effect, and also produces inhibitory effects such as enterprise resource occupation effect and employment delay effect. This paper extends the research on the impact of industrial technological advances on the labour employment demand of companies, and provides empirical evidence and policy insights for rationally arranging industrial structural transformation and labour employment decisions of companies in the context of 'stable employment'.

## 1. Research background

The Law of the People's Republic of China on Scientific and Technological Advance (2022) proposes to promote technological advance in order to achieve high-quality development of

**Funding:** The author(s) received no specific funding for this work.

**Competing interests:** The authors have declared that no competing interests exist.

the Chinese economy. Technological advance is an evolutionary process carried out by human beings by transforming or innovating the original technology, or developing new technologies to replace the old ones, and it is an evolutionary process carried out by technology to achieve certain goals [1]. Structural growth theory suggests that one of the important causes of industrial structural change is labour productivity improvement originating from innovation and technological advance [2]. Technological advance is the main driving force and core element to accelerate industrial upgrading, and is the core driving force for economic growth. A series of new technologies enter the industrial level, undergo industrialisation to become "technology in industry", and then spread and popularise in industry to become industrial technology [3]. Based on the above evolutionary process, industrial technology comes from the combination of various types of technology at the industrial level, which is the evolutionary process of technology to achieve certain goals at the industrial level, the technology space boundary and the share of different types of technology in the technology space in the intra-industry and inter-industry optimisation, combination, and the efficiency of factor utilisation at the industrial level is improved, resulting in the formation of technological advance in industry.

In recent years, as China's industrial structure continues to upgrade, technological advance in industry has increased to become the norm in the economy and society. Technological advance in industry has given rise to a number of social and economic problems, among which employment has become one of the major issues of widespread concern in both academic and practical circles. Throughout the history of world economic development in the past two hundred years, several rounds of industrial transformation and upgrading, such as steam engines, railways, electricity, computers, and intelligence, have experienced wave after wave of technological advance in industry. In the evolution of technological advance in industry, the demand for and structure of corporate labour employment tend to fluctuate considerably, and even phenomena such as social employment opportunities and waves of unemployment with huge impacts have occurred. As early as 1930, Keynes made the speculation that mankind would face technological unemployment in the next 90 years [4]. In a report released by McKinsey Global Institute, it was stated that with the advancement of technology, 75 million to 375 million people will be re-employed and learn new skills globally by 2030, and in terms of numbers, China will face the largest change in employment, with an estimated 12 million to 102 million Chinese needing to be re-employed [5] The rapid development of technology has brought unprecedented and huge challenges to human work [6]. Influenced by the major adjustment of economic structure, changes in labour force structure, imperfect market employment mechanism, and economic and trade friction between China and the United States, China is currently facing a severe employment situation.Data show that from 2010–2019, China's employment rate of people over 15 years of age has continued to decline (Fig 1). China is a large country in terms of population and labour force, with a working-age population of 880 million people aged 16–59 at the end of 2021, which will remain above 850 million during the Fourteenth Five-Year Plan period, exceeding the combined population of Europe. Since the 18th CPC National Congress, China has deeply implemented the strategy of giving priority to employment, and made employment the top priority of the "six stabilisers" and "six guarantors", and the 2019 Government Work Report for the first time puts the policy of giving priority to employment at the level of macro policies, and gives employment a more prominent position in all government work, Giving employment a more prominent position.The government work report of the 2023 two sessions further implements and details the employment priority policy and promotes high-quality full employment.

Companies are the micro-carriers that undertake social employment and solve social employment problems mainly through labour hiring [7–10]. Studies have been conducted to explore the influencing factors of labour employment in companies from various aspects; in

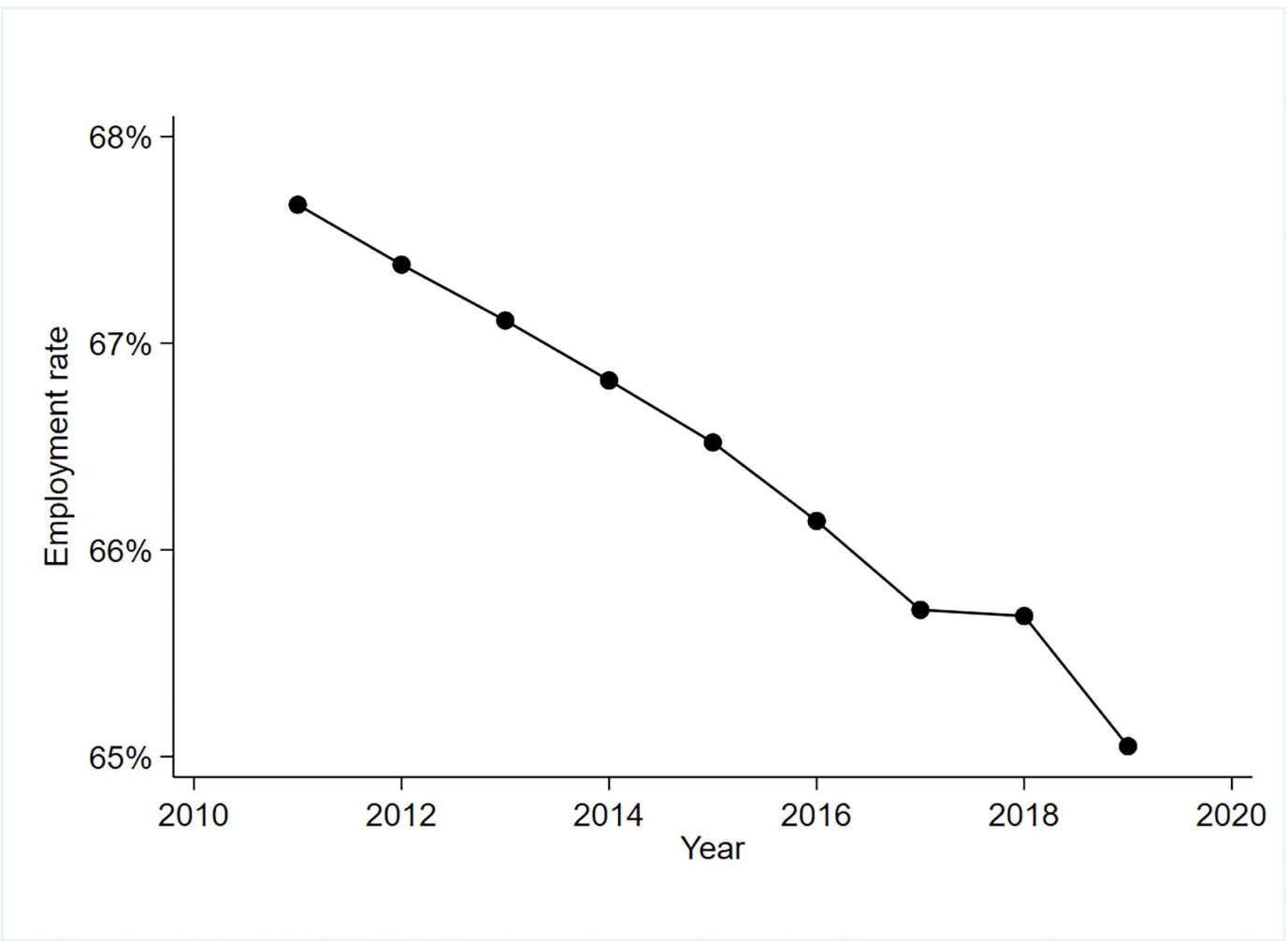

**Fig 1. Employment rate of the population aged 15+, China, 2011–2019.** Source: World Bank website.

terms of macro policies, tax reduction incentives increase labour employment in upstream and downstream companies through supply chain transmission [11, 12], and tax incentives for accelerated depreciation mainly affect labour employment through the "substitution effect" and "output effect " affecting labour employment [13–15], and the increase in social insurance contributions reduces firms' liquidity, thus lowering the growth rate of firms' labour employment [16]. In terms of market environment, the decline of market risk premium enhances the growth rate of corporate labour employment at a certain time in the future and starts to reverse after a period of time, and there is a time lag in employment [17], digital finance [18], local government debt financing [19], and patent pledging [20] affects corporate labour hiring by affecting corporate financing constraints labour hiring, and financing constraints have a significant negative impact on corporate hiring demand [21]. In terms of the structure of corporate labour employment, The upgrading of information technology allows more and more work to be done by artificial intelligence, thus reducing the need for existing labour and increasing the need for skilled personnel [22, 23], the recognition of high-tech companies significantly promotes the increase of corporate labour employment [24].

Previous literature is mainly about macro-level technological advances as well as micro-level technological advances, while there is an extreme lack of relevant research on the industrial technology level. In practice, industrial technological advances have caused changes in the demand, structure and quality of labour hiring in companies, and some scholars [25] have begun to pay attention to the possible impact of specific technologies such as robotics applications, artificial intelligence, automation, digital transformation and other technologies applied in companies on the labour market in China, but the impact of industrial technologies in a broad sense on the labour hiring decisions of companies and their mechanisms of action has not yet been carried out. This paper provides an in-depth analysis of the evolutionary process of industrial technology, and refines and summarizes the formation, scope and nature of industrial technological advances, based on which it theoretically extrapolates and empirically examines its impact mechanism on company labour hiring, expanding the study of the microeconomic consequences of industrial technology.

Industrial technology suffers from a nonlinear evolutionary process in which technologies evolve from discrete distributions that do not interfere with each other to highly agglomerative network structures that evolve synergistically [26]. The fundamental nature of industrial technology is complexity [3], and industrial technology complexity specifically exhibits the characteristics of non-linearity, unpredictability, divisibility and systemicity [27]. The relevant literature mainly focuses on the fragmentation of 'theoretical elaboration', 'measurement' [28] and 'influencing factors', and has not yet formed a more systematic theoretical study. A more systematic theoretical system has not yet been formed. Previous research on specific technological advances such as artificial intelligence, automation, digital transformation, etc. has not examined the influence mechanism from the perspective of these characteristic facts. In this paper, the analysis of labour employment based on the characteristics of industrial technological advances is obviously more in line with the logic of reality and the paradigm of theoretical deduction, and it is also an enrichment and supplementation of the original industrial technology theory.

In the current context of China's industrial structure upgrading and employment stability, industrial technology advancement will inevitably trigger the adjustment of corporate labour employment decision-making, leading to changes in corporate labour hiring demand, hiring structure, and hirer remuneration. However, there is an extreme lack of existing research, with only a few articles on the impact of robotics applications [29] and artificial intelligence, etc. on China's labour market, and few scholars have explored the impact of industrial technological advances on corporate labour hiring and its mechanism of action. Based on the above analysis, this paper takes A-share listed companies from 2011 to 2020 as the research object to empirically test the impact of technological advance in industry on corporate labour hiring decisions.

The marginal contribution of this paper lies in the following aspects: Firstly, it empirically examines the relationship between technological advance in industry and company labour employment, provides new evidence for the "inhibitory effect" and "incentive effect" of employment from the perspective of technological advance in industry, and expands the research on the microeconomic consequences of technological advance in industry. Secondly, it enriches the research on the influencing factors of labour employment in companies, and provides new perspectives for alleviating the current employment pressure in China and promoting high-quality full employment. Third, it explores the influence mechanism of technological advance in industry on company labour employment, and on this basis, tests the degree of influence of technological advance in industry on company labour employment under different levels of internal and external environments.

## 2. Theoretical analysis and hypothesis development

Industrial technology is the existential form of technology evolution to the industrial level [30]. Some scholars have used community succession dynamics to fit the various evolutionary stages of industrial technology, and found that industrial technology exists a nonlinear evolutionary process in which technology is distributed discretely from mutual non-interference, and can progress to a highly agglomerative network structure of synergistic evolution [27–29, 31, 32]. In the evolutionary process, technological advance in industry is highly dependent on the environment; there is structural diversity and hierarchical diversity in technological advance in industry; the timeliness and value of technological advance in industry decreases over time; there is intense competition or learning and cooperation between various technologies within the industry, and the relationship between technologies also changes over time. From the evolution process, it can be seen that industrial technological progress has the characteristics of complexity, divisibility, systematicity and market selectivity. ①technological advance in industry has complexity. The process of industrial technological progress is accompanied by such complexity features as nonlinearity, interaction, trajectory, emergence and system embedded system, which is specifically manifested in the structural diversity and hierarchical diversity of industrial technology [33], and the more complex industrial technology, the more the need for companies to increase the demand for labour employment in complex technology to adapt to it; and at the same time may also reduce or eliminate the demand for the original simple technology jobs; ②Industrial technological progress is divisible and systematic.technological advance in industry makes the division of labour continue to refine and deepen, driven by factor decomposition, craft decomposition, process decomposition, industrial technology in a link can even be divided into a number of small segments, a large number of intermediate products, technological advance in industry makes industrial technology divisibility is greatly enhanced, and promote vertical separation. Technological advance in industry implies a series of new and interrelated technologies, which firms in the chain need to apply and to substitute, co-operate and complement the technologies of other firms; ③Technological advance in industry is market-selective. Technological advance in industry is the result of market selection and elimination of technology, the main criterion of market selection is based on cost-benefit judgement, otherwise the technology may not enter the industrial level. The market mechanism selects the technology to form the new technology in the industry, the new technology through the industry internal promotion and popularisation, often can bring more profit and controllable risk to the company and become the industrial technology, industrial technology means that the maturity and uncertainty of technology application is weakened, and the cost and risk of new technology application by companies are subsequently reduced, which enhances companies' investment in resources such as capital expenditure on industrial technology, R&D and high-tech application. In addition, market selectivity leads to the diffusion of industrial technology knowledge in the industry and guides the rational flow of resources among companies.

Whether the relationship between technological advances in industry and firms' demand for labour employment manifests itself as an incentive or a disincentive is related to the very nature of technological advances in industry.

### 2.1 Incentive effects of technological advances in industry on firms' labour hiring decisions

**2.1.1 Technological advance in industry triggers the expansion of technological investment and increases company labour employment.** Technological advance in industry, as an important social and technological phenomenon and institutional phenomenon, forms the

operational basis, rules and field of microcompany technological research and development and application, forcing companies in industry to invest in new technological research and development, application and other investment behaviours, etc., and the expansion of investment tends to induce the decision-making that increases the demand for company labour employment. In particular, in a certain period of time and under certain resource constraints, the market selectivity of technological advance in industry determines the flow of resources to companies adapting to technological advance in industry, which can obtain higher production efficiency and lower operating costs, and the demand for company labour employment increases. On the one hand, technological advance in industry is conducive to improving production efficiency, reducing the price of factors of production, so that companies save production and operation costs, profitability increased, and actively expand the scale of production and operation, thus companies increase the demand for labour employment [34]. Technological advance in industry will not only improve production efficiency, long-term capital accumulation will also lead to technological advance again to improve productivity, and directly increase the demand for labour in the segment complementary to the new technology [35]; on the other hand, technological advance in industry leads to a decline in the cost of production and operation of companies, which leads to a decrease in the price of goods and services affected by technological advance in industry, and a rise in the real income of consumers. Through the "income effect", consumers' consumption demand for the product increases, and the resulting expansion of output motivates companies to further expand the scale of production and operation, and companies will increase the demand for labour employment in various segments [36].

**2.1.2 Technological advance in industry creates demand for technical jobs.** With technological advance in industry, the higher the complexity means the more new technology, the new technology induces consumers to increase their demand for technological additions to their products, motivates companies in related industries to expand the technological content of their products, and also creates demand for new employment in companies employing skilled labour. The complexity of technological advance in industry brings about an increase in the demand for skilled jobs, for example, in jobs where technology has a comparative advantage over labour, potentially creating jobs where labour has a comparative advantage over technology, Acemoglu and Restrepo point out that the job creation effect explains the 1980–2010 employment growth by about half in the United States [22], and Gregory & Salomons argue that the demand for new businesses, new models and new jobs generated tends to outnumber the number of jobs replaced [37]. Technological advances in industry want to give full play to the benefits of output and adapt to the different production environments of companies, which requires labour with a certain level of skill to collaborate with each other, which in turn gives rise to labour positions with higher skill requirements, and companies train the labour force of the original positions or recruit labour with a high level of skill. These jobs often require higher human capital, and firms pay more attention to the specific skills and accumulated experience of their employees, which further increases the cost of terminating highly skilled employees [38].

**2.1.3 The level of technological advance in industry in question not only creates demand for skilled labour employment in companies, but also generates an increase in the number of jobs in service industries associated with it.** The divisibility and systematic nature of technological advance in industry means that there is an increase in the demand for jobs that are complementary and supplementary to the new technology. For example, with the development of computer science, the production and use of computers has become more complex, creating a large number of technical job demands for labour such as software development, network security and maintenance, programmers, etc. Especially with the increase in the

remuneration of technical jobs, there is a further increase in the number of service jobs such as commerce, tourism and entertainment, and the satisfaction of these demands in turn incentivises a range of hiring decisions for skilled jobs, generating a service job complementarity effect. Dauth et al.'s study, based on German data, also found that new technological applications brought about an increase in labour hiring of labourers in the service sector [39].

**Hypothesis 1a:** Technological advances in industry may incentivise firms' demand for labour hire.

## 2.2 Technological advances in industry have a dampening effect on labour demand

**2.2.1 Technological advance in industry leads to the occupation of company resources and reduces the demand for labour hiring.** The higher the complexity of technological advance in industry, the higher the number of possible states exists [40], which contains the more diverse set of technological investment opportunities. The investment opportunity set formed by the complexity of industrial technology prompts companies to develop new products, improve production processes or change product quality and structure, and improve tools, equipment, instruments, devices, inspection methods, etc. Firms can engage in industrial technology investments from multiple entry points. A series of technology investments may have an "encroachment effect" on firms' resources [41]. The more complex the technology link is, the more internal resources such as people and materials are required, crowding out resources that should be used for other businesses such as operations, and suppressing the demand for labour employment in the company. In particular, companies will also face high risks and long cycles when investing in new technologies, which also means that a large amount of capital may be over-expended by the company, which will also affect the company's future investment in its sustainable business capacity and inhibit the demand for labour employment.

**2.2.2 Technological advance in industry has led to the replacement of old jobs.** The market selectivity of technological advance in industry implies the elimination mechanism, the new technology replaces the old technology, and many of the original jobs related to the old technology are replaced. First of all, when technology has a comparative advantage over labor, companies in order to save costs and improve productivity, labor will be replaced by technology, that is, the substitution effect.Acemoglu and Restrepo further pointed out that the emergence of new technology not only affects the replacement of labor by the current technology, but also leads to the allocation of the new task to the lower price of capital rather than higher price of labor, resulting in the "substitution effect" creating a 'displacement effect' that further deepens the negative impact on labour [42]. In recent years, with the rise of big data and deep learning, some positions that are traditionally regarded as irreplaceable by technological progress, such as manual customer service and accounting, may also be replaced by new technologies, and the scope of routine tasks is getting wider and wider, so that every task has the possibility of being replaced by technology in the future, and everyone faces the risk of becoming a labourer for routine tasks. Secondly, technological advance in industry will also exacerbate the Matthew effect, with a few large companies taking the lead in completing the technological substitution of part of the labour force by virtue of their scale advantage and strong financial strength, squeezing the living space of small and medium-sized companies and indirectly substituting part of the labour force of small and medium-sized companies. Finally, with the continuous progress of industrial technology, the production costs of

companies continue to reduce, production efficiency continues to improve, so companies gradually increase the relevant capital investment, capital deepening, the introduction of more advanced technology to replace the corresponding positions of the labour force. In the short term, the complementarity between capital and labour makes the accumulation of capital accompanied by an increase in the demand for employment, but in the process of long-term development, the substitution of capital for labour gradually appears, and the absorbing effect of capital on employment is weakened [43]. For example, at the beginning of the industrial revolution in Britain, the great development of the machine industry enabled the simple operation of workers to replace the skilled craftsmen in the original craft workshops or workshops, and a large number of skilled craftsmen lost their jobs through a series of specialised assembly line technologies replacing the crafts technologies. Technological advance in industry mainly simplified the established production technology process, breaking down complex labour into simple labour on the assembly line, and technological advance in industry and the demand for company labour employment mainly showed a suppressive relationship. Each generation of innovations in the early stages of computer software has made operation easier and simpler, increasing the labour efficiency and labour demand of ordinary users, and thus reducing the demand for computer specialists.

**Hypothesis 1b:** Technological advances in industry may dampen the demand for corporate labour hire.

## 3. Research design

### 3.1 Sample selection and data sources

This paper selects the A-share listed companies in Shanghai and Shenzhen from 2008 to 2022 as the initial research sample, excludes the financial companies and ST companies, excludes the samples with missing variables, and excludes the samples with the number of active employees less than 10, and finally obtains 40,601 observations. The data are mainly obtained from the Cathay Pacific (CSMAR) database and the China Statistical Yearbook. In this paper, all continuous variables are indented at the top and bottom 1%.

### 3.2 Variable setting

**3.2.1 Explained variables.** Firm labour employment (lnNumber). In this paper, we measure the natural logarithm of the number of active employees in a firm.

**3.2.2 Explanatory variables.** Technological advance in industry (tfpc). Existing studies have mainly used total factor productivity (TFP) to measure the level of technological progress, and TFP measurement mainly includes growth accounting methods [42–44], nonparametric methods [45–47], and parametric methods [48, 49]. In this paper, we measure the level of technological advance in industry by using the tfp index of the industry measured by data envelopment analysis (DEA).

**3.2.3 Control variables.** This paper controls for firm-level and regional-level factors that may have an impact on firm labour employment. Firm-level control variables include: firm size (Size), gearing ratio (Lev), cash flow ratio(Cashflow), firm growth (Growth), dual employment (Dual), and ownership of companies(SOE). Control variables at the regional level include: gross regional product (lnGdp), and secondary industry share (is). In addition, the paper controls for year fixed effects and city fixed effects. The specific definitions of each variable are shown in Table 1.

**Table 1. Variable definition.**

| Variable Name | Notation | Variable Definition |
|---|---|---|
| Company labour employment | *lnNumber* | Natural logarithm of the number of active employees in the company |
| Technological advance in industry | *tfp* | Industry total factor productivity |
| Company size | *Size* | Natural logarithm of total assets |
| Gearing | *Lev* | Total liabilities/Total assets |
| Cash flow ratio | *Cashflow* | Net cash flows from operating activities/current liabilities |
| Company growth | *Growth* | Operating income for the year/Operating income for the previous year-1 |
| Dual employment | *Dual* | The chairman of the board and the general manager are the same person as l, otherwise 0 |
| Ownership of companies | *SOE* | State-owned companies take 1, otherwise it is 0 |
| Gross regional product | *lnGdp* | Natural logarithm of gross regional product |
| Percentage of secondary sector | *is* | Ratio of secondary sector output to GDP |
| City | *city* | City dummy variables |
| Year | *year* | Year dummy variable with a sample interval of 2008–2022 |

## 3.3 Establishment of model

$$\ln Number_{it} = \alpha_0 + \alpha_1 tfp_{ct} + \sum control + \sum city + \sum year + \varepsilon_{ict} \tag{1}$$

In the above Eq (1), lnNumber$_{it}$ is the labour employment of the company i in year t, tfpct represents the level of technological advance in industry, control is the set of control variables, city and year denote the city and year dummy variables; εict is the random error term.

## 4. Empirical testing

### 4.1 Descriptive statistics

Table 2 presents the descriptive statistical characteristics. The maximum value of the number of employees in the company (Number) is 65159, the minimum value is 82, and the standard

**Table 2. Descriptive statistics.**

| Variant | Observations | Mean | Standard deviation | Minimum | Median | Maximum |
|---|---|---|---|---|---|---|
| *Number* | 40601 | 4778.3704 | 9257.5992 | 82 | 1852 | 65159 |
| *lnNumber* | 40601 | 7.5954 | 1.2694 | 4.4067 | 7.5240 | 11.0846 |
| *DEA* | 40601 | 0.6844 | 0.1857 | 0.2626 | 0.6921 | 1.1395 |
| *Size* | 40601 | 22.1163 | 1.3459 | 10.8422 | 21.9220 | 28.6365 |
| *Lev* | 40601 | 0.4229 | 0.2120 | 0.0505 | 0.4128 | 0.9472 |
| *Cashflow* | 40601 | 0.0472 | 0.0710 | -0.1720 | 0.0465 | 0.2495 |
| *Growth* | 40601 | 0.1750 | 0.4274 | -0.5744 | 0.1093 | 2.7799 |
| *Dual* | 40601 | 0.2879 | 0.4528 | 0 | 0 | 1 |
| *SOE* | 40601 | 0.3539 | 0.4782 | 0 | 0 | 1 |
| *lnGdp* | 40601 | 10.5169 | 0.7981 | 7.9570 | 10.5904 | 11.7685 |
| *is* | 40601 | 41.3692 | 9.3684 | 15.9000 | 42.7000 | 56.3300 |

Source: Cathay Pacific database and manual collation.

deviation is 9257.60, indicating that the data on labour employment in the company varies a lot, so the natural logarithm is taken for the labour employment in the company in order to attenuate the heteroscedasticity of the data when conducting the empirical test. The tfp measured with the DEA square varies widely, with the maximum value about 4 times the minimum value, indicating that the level of technological advance in industry varies widely among companies.

## 4.2 Baseline regression analysis

In this paper, we use ordinary least squares (OLS) for panel data regression. Column (1) of Table 3 shows the results of the univariate test, column (2) adds control variables, and the coefficients of the key explanatory variables, tfp, are all significantly negative at the 1% level. Column (3) is the results of the test that adds the city and time fixed effects, and the coefficients of the explanatory variables, tfp, decrease, which suggests that some of the city-level characteristics that do not change over time are omitted variables that affect firm labor hiring, and not controlling for these variables would overestimate the role of industrial technological progress. The regression results indicate that the higher the level of technological advance in industry, companies will reduce labor hiring, indicating that the technological advance in industry generally suppresses the demand for labor hiring by companies, and hypothesis 1b is verified.

**Table 3. Basic regression results.**

| Variant | (1) | (2) | (3) |
|---|---|---|---|
| | lnNumber | lnNumber | lnNumber |
| tfp | -0.5996*** | -0.3877*** | -0.6139*** |
| | (-17.7405) | (-16.5350) | (-24.2077) |
| Size | | 0.6795*** | 0.6921*** |
| | | (181.4567) | (184.5805) |
| Lev | | 0.1937*** | 0.1563*** |
| | | (8.3639) | (6.8358) |
| Cashflow | | 2.0964*** | 1.9170*** |
| | | (34.4632) | (32.4300) |
| Growth | | -0.0942*** | -0.0978*** |
| | | (-9.5349) | (-10.3303) |
| Dual | | 0.0380*** | 0.0456*** |
| | | (3.8590) | (4.7892) |
| SOE | | -0.0041 | -0.0507*** |
| | | (-0.3988) | (-4.8531) |
| lnGdp | | -0.0681*** | -0.0106 |
| | | (-12.0350) | (-0.2217) |
| is | | 0.0163*** | -0.0021 |
| | | (34.9678) | (-0.9667) |
| _cons | 8.0057*** | -7.3000*** | -6.7991*** |
| | (334.0386) | (-73.5699) | (-15.6164) |
| City fixed effect | No | No | Yes |
| Year fixed effect | No | No | Yes |
| N | 40601 | 40601 | 40601 |
| p>|t| | p<0.001 | p<0.001 | p<0.001 |
| Adj. R2 | 0.0077 | 0.5553 | 0.6076 |

## 4.3 exogenous policy shock

Industrial technological advancement is often associated with the degree of regional marketisation and openness, and new state-level zones have a driving effect on regional economic growth, belong to comprehensive economic functional zones, and are more likely to realise industrial agglomeration and technological innovation, and promote industrial technological advancement in the region. In order to more robustly assess whether industrial technological advances can influence the decision on the number of workers employed by companies, this paper adopts the establishment of state-level new zones as an exogenous shock policy, and evaluates it with the differences-in-Differences (DID) method.19 state-level new zones were established one after another in 1992, 2006, 2010, 2011, 2012, 2014, 2015, 2016 and 2017 were approved in batches.In order to overcome the endogeneity problem caused by the non-randomness of the cities covered by the state-level new zones, this paper adopts PSM-DID to construct the model (2):

$$\ln Number_{it} = \beta_0 + \beta_1 \mathrm{did}_t + \sum control + \sum \mathrm{city} + \sum year + \varepsilon_{ict} \qquad (2)$$

Where $\ln Number_{it}$ is the explanatory variable, the number of companies employing labour; $\mathrm{did}_t$ is the policy variable, i.e. the product of the treatment group dummy variable and the policy implementation time dummy variable. The treatment group dummy variable: cities covered by national-level new zones during the sample period are assigned a value of 1, and those not covered are assigned a value of 0. The national-level new zone policy implementation time dummy variable: cities are assigned a value of 0 before approval, and a value of 1 afterwards.

After regressing the model (2) through the parallel trend test and placebo test, the results in Table 4 show that the regression coefficient is significantly negative, and the establishment of the new state-level zones brings about the advancement of industrial technology in the region, and the advancement of industrial technology has an inhibitory effect on the number of companies' labour, which in turn reduces the number of companies' labour.

## 4.4 Robustness tests

**4.4.1 Key variable substitution.** In this paper, we use the stochastic frontier analysis (SFA) method and a generalised method of moments (GMM) computed by Blundell and Bond (1998) to measure technological advances in industries [50], and the results are shown in columns (1) and (2) of Table 5, and the findings are robust. Then, this paper further uses the LP and OP methods to calculate the total factor productivity of enterprises and takes the industry mean as a proxy variable for industrial technological progress, and the results, as shown in

**Table 4. Exogenous shock test.**

|  | (1) *lnNumber* |
|---|---|
| did | -0.0760** |
|  | (-2.4122) |
| _cons | -6.3175*** |
|  | (-8.8663) |
| *Control* | Yes |
| *City* | Yes |
| *Year* | Yes |
| N | 14222 |
| *p>\|t\|* | 0.016 |
| Adj R-sq | 0.5492 |

**Table 5. Key variable substitution.**

|  | (1) | (2) | (3) | (4) |
|---|---|---|---|---|
|  | *lnNumber* | *lnNumber* | *lnNumber* | *lnNumber* |
| SFA | -0.6509*** |  |  |  |
|  | (-26.9343) |  |  |  |
| *GMM* |  | -0.5499*** |  |  |
|  |  | (-22.5842) |  |  |
| *LP* |  |  | -0.2220*** |  |
|  |  |  | (-24.5799) |  |
| *OP* |  |  |  | -0.3188*** |
|  |  |  |  | (-44.9943) |
| *_cons* | -6.6107*** | -6.7123*** | -5.9925*** | -8.7625*** |
|  | (-15.2002) | (-15.3913) | (-13.6626) | (-19.7455) |
| *Control* | Yes | Yes | Yes | Yes |
| *Industry* | Yes | Yes | Yes | Yes |
| *Year* | Yes | Yes | Yes | Yes |
| N | 40601 | 40601 | 40584 | 36431 |
| $p > |t|$ | p<0.001 | p<0.001 | p<0.001 | p<0.001 |
| Adj R-sq | 0.6089 | 0.6068 | 0.6074 | 0.6218 |

columns (5) and (6) of Table 5, show that the regression coefficients of LP and OP are significantly negative at the 5 per cent level, a result that further supports the findings of this paper.

**4.4.2 Placebo test.** Consider the possibility that the correlation in this paper's benchmark regression is simply a placebo effect, i.e., a negative correlation between technological advances in industry and corporate labour hiring in the data of this paper due to unobserved limitations in the design of the study, which is in fact not related to the inhibitory effect of technological advances in industry on corporate labour hiring. In order to ensure the robustness of the findings, this possibility is ruled out in this paper using a placebo test. This paper extracts the values of tfp variables from all firm-year observations in the sample data set [51], these values are then randomly assigned to each "company-year" observation one by one, and finally the model (1) is regressed again. The regression results are shown in column (1) of Table 6, where the coefficient on tfp is insignificant and significantly different from the results of the baseline regression, implying that the placebo effect does not exist, again validating the robustness of the findings.

**4.4.3 Instrumental variables test.** In this paper, the distance from each province and region to the nearest port (IV) is chosen as an instrumental variable, and the distance from each province and region to the port reflects the front-end, middle-end and back-end locality of the technological division of labour in the industry chain with differences in the market environment, and different technological positions will depend on differences in the market environment. Therefore, foreign market proximity forms an important external environment for industrial technological progress, can screen local industrial technology with high correlation with the formation of industrial technological progress, but does not directly affect enterprise labour employment, and is not correlated with enterprise labour employment and the residual term of the model, which can be used as an instrumental variable for industrial technological progress. The instrumental variable Cragg-Donald Wald F-statistic is greater than the 15% level critical value, indicating that there is no problem of weak instrumental variables, and the p-value is significant at the 1% level in the non-identifiability test and the endogeneity

**Table 6. Robustness tests.**

| | (1) | (2) | (3) | (4) | (5) | (6) |
|---|---|---|---|---|---|---|
| | *lnNumber* | *tfp* | *lnNumber* | *lnNumber* | *lnNumber* | *lnNumber* |
| | *Placebo test* | *Instrumental variable test* | | *Replacement sample interval* | *2% and 98% indentation* | *5% and 95% indentation* |
| tfp | 0.0149 | | -6.5982*** | -0.5874*** | -0.6380*** | -0.5692*** |
| | (0.6923) | | (-7.7893) | (-21.4156) | (-25.8904) | (-24.6161) |
| *IV* | | -0.0064*** | | | | |
| | | (-9.2318) | | | | |
| _cons | -7.3638*** | 0.8961*** | -3.1376*** | -6.8132*** | -7.7557*** | -8.0385*** |
| | (-16.8059) | (38.2367) | (-4.6335) | (-15.0932) | (-18.9440) | (-22.7872) |
| *Control* | Yes | Yes | Yes | Yes | Yes | Yes |
| *City* | Yes | No | No | Yes | Yes | Yes |
| *Year* | Yes | Yes | Yes | Yes | Yes | Yes |
| N | 40601 | 40601 | 40601 | 35405 | 40601 | 40601 |
| *p>|t|* | 0.489 | p<0.001 | | p<0.001 | p<0.001 | p<0.001 |
| Adj R-sq | 0.6019 | 0.4407 | | 0.6065 | 0.5920 | 0.5821 |

test, i.e. the original hypothesis is rejected. Column (3) of Table 6 presents the results of the second stage regression, where the coefficients of the key explanatory variables are all significantly negative at the 1 per cent level. This indicates that the results remain robust after controlling for possible endogeneity issues.

**4.4.4 Other robustness tests.** First, in order to exclude the impact of the 2015 stock market crash on the empirical results of this paper, we exclude the samples of 2015 and 2016 and re-run the main regression, and the results are shown in Table 6. The data show that the effect of technological advance in industry on corporate labour employment is still significantly negative, indicating that the results of this paper are more robust. Second, to ensure that the test results are not driven by outliers, continuous variables are further subjected to 2% and 98% shrinkage and 5% and 95% shrinkage respectively. None of the robustness test results are substantially altered.

# 5. Heterogeneity test

Technology has a comparative advantage over manpower in conventional tasks, while manpower has a comparative advantage over technology in non-conventional tasks. The elasticity of substitution of technological progress for different tasks varies, with a negative "substitution effect" and a positive "complementarity effect". On the one hand, low-skilled labour is more likely to be replaced by technology in technological advance in industry than high-skilled labour, and technology has a comparative advantage over manpower, so firms reduce their investment in low-skilled labour. High-skilled labour possesses unique knowledge and skills relative to low-skilled labour, and firms tend to increase their investment in high-skilled labour in order to reduce training costs and directly acquire this unique knowledge and skills. On the other hand, with the technological advance in industry, under the automated production environment, some low-skilled jobs have been replaced by technology, while also creating jobs in which manpower has a greater comparative advantage over technology, expanding labour demand.

Technological progress has produced a labour "substitution effect", mainly in the manufacturing sector, so that companies will reduce the employment of production personnel. As far as financial positions are concerned, financial management is a central part of company

**Table 7. Technological advance in industry and the educational structure of employees in companies.**

|  | (1) | (2) | (3) |
|---|---|---|---|
|  | **Percentage of high education level** | **Percentage of low education levels** | **Percentage of other educational levels** |
| tfp | 19.3503*** | -13.7259*** | -13.8877*** |
|  | (23.9370) | (-15.7720) | (-12.3135) |
| _cons | 34.2288*** | 113.8872*** | 50.7683 |
|  | (3.0409) | (5.5146) | (1.6181) |
| Control | Yes | Yes | Yes |
| City | Yes | Yes | Yes |
| Year | Yes | Yes | Yes |
| N | 25727 | 18700 | 18988 |
| p>|t| | p<0.001 | p<0.001 | p<0.001 |
| Adj R-sq | 0.2743 | 0.2774 | 0.1972 |

management, and the demand for high-end financial personnel has increased as companies seek to achieve digital transformation of their financial systems to better support management and assist in operations.

This paper classifies labour employment in companies on the basis of the educational level of their employees and their positions. Bachelor's degree and postgraduate and above are defined as high education level, and specialist and high school and below are defined as low education level; according to the employee position, they are divided into production personnel, financial personnel and technical personnel. Table 7 grouped according to the educational level of employees, the test results show that technological advance in industry mainly reduces the labor employment for the share of employees with low educational level and increases the labor employment for the share of employees with high educational level. Table 8 grouped by employee position, the test results show that technological advance in industry mainly reduces the labor employment of production staff and increases the labor employment of finance staff.

## 6. Mechanism test

### 6.1 Incentive effects: A test of investment expansion effects and chain transmission effects

**6.1.1 Technology investment expansion effect.** The previous theoretical analysis suggests that industrial technological advances generate investment expansion effects and increase

**Table 8. Technological advance in industry and employee job structure in companies.**

|  | (1) | (2) | (3) | (4) | (5) |
|---|---|---|---|---|---|
|  | **Percentage of production staff** | **Percentage of finance staff** | **Percentage of technical staff** | **Percentage of sales staff** | **Percentage of other personnel** |
| tfp | -16.8833*** | 1.5446*** | 8.3563*** | 7.3878*** | 8.7936*** |
|  | (-23.4264) | (15.9129) | (13.9346) | (13.9715) | (16.2296) |
| _cons | 45.6300*** | 9.3946*** | 17.3446 | 18.5659 | -8.8030 |
|  | (2.6125) | (3.5570) | (1.1686) | (1.4175) | (-0.6525) |
| Control | Yes | Yes | Yes | Yes | Yes |
| City | Yes | Yes | Yes | Yes | Yes |
| Year | Yes | Yes | Yes | Yes | Yes |
| N | -16.8833*** | 1.5446*** | 8.3563*** | 7.3878*** | 8.7936*** |
| p>|t| | p<0.001 | p<0.001 | p<0.001 | p<0.001 | p<0.001 |
| Adj R-sq | 0.2258 | 0.1439 | 0.1701 | 0.1154 | 0.1061 |

**Table 9. Technology investment expansion effects and chain transmission effects.**

|  | (1) | (2) | (3) | (4) | (5) |
|---|---|---|---|---|---|
|  | *lnincome* | *lnNumber* | *lnNumber* | *lnNumber* | *lnNumber* |
|  | *Technology investment expansion effect* | *Upstream pass-through effect (industry i is upstream)* | | *Downstream effects (industry i is downstream)* | |
| tfp | 0.0039*** | | -0.9529*** | | -0.8041*** |
|  | (4.9987) | | (-23.7675) | | (-24.5487) |
| *Upstream$_{ct}$* | | 0.2553*** | 0.5697*** | | |
|  | | (3.1918) | (7.1246) | | |
| *Dowmstream$_{ct}$* | | | | 0.5668*** | 0.7941*** |
|  | | | | (5.6408) | (7.9499) |
| *_cons* | 0.0493*** | -5.6118*** | -4.8197*** | -8.0124*** | -7.2986*** |
|  | (3.1417) | (-8.0640) | (-7.0170) | (-14.9749) | (-13.7597) |
| *Control* | Yes | Yes | Yes | Yes | Yes |
| *City* | Yes | Yes | Yes | Yes | Yes |
| *Year* | Yes | Yes | Yes | Yes | Yes |
| N | 25605 | 19762 | 19762 | 29630 | 29630 |
| Adj R-sq | 0.1877 | 0.5353 | 0.5484 | 0.6070 | 0.6150 |

company labour employment. This paper uses the ratio of company R&D investment to total assets to measure the scale of enterprise investment, Table 9, column (1) column (1) of the estimation results of industrial technological advances on the scale of enterprise investment, industrial technological advances in the 1 per cent level is significantly positive, technological investment expansion effect exists.

**6.1.2 Industry chain transmission effect.** The theoretical analysis in the previous section suggests that through industry chain transmission, labour hiring jobs of industrial technological advances can be transmitted to companies upstream and downstream of the industry chain. This paper identifies the relatively important downstream and upstream industries of each industry according to China's input-output table (2010) as follows: for industry i, if more than 1 per cent of the output of industry i is put into another industry k for use, and the proportion of the output of industry k that is put into industry i is not more than 1 per cent, industry k is considered to be a relatively important downstream industry of industry j, and industry j is a relatively important upstream industry of industry k.

$$\ln Number_{it} = \gamma_0 + \gamma^{\text{own}} tfp_{ct} + \gamma^{up} Upstream_{ct} / \gamma^{down} Downstream_{ct}$$
$$+ \sum control + \sum city + \sum year + \varepsilon_{ict}$$

(3)

The $tfp_{ct}$ coefficient $\gamma^{\text{own}}$ reflects the impact of industrial technological advances on the labour employment of companies in industry i. The $Upstream_{ct}$ coefficient $\gamma^{up}$ measures the impact of industrial technological advances on the labour employment of companies in the downstream industries of industry i, which is known as the 'upstream transmission effect', while the $Downstream_{ct}$ coefficient $\gamma^{down}$ reflects the impact of industrial technological advances in the upstream industries of industry i on the labour demand of companies in industry i, which is known as the 'downstream transmission effect'. The coefficient γup of Downstreamct reflects the impact of industrial technological advances in the upstream industries of industry i on the labour demand of companies in industry i, which is called the 'downstream transmission effect'. In this paper, drawing on Acemoglu et al. (2016), $Upstream_{ct}$ and

Downstream$_{ct}$ are constructed as follows:

$$Upstream_{ct} = \text{tfp}_{ct} * \sum\nolimits_{k}(output\%_{i \to k}^{2010})$$ (4)

$$Downstream_{ct} = \text{tfp}_{ct} * \sum\nolimits_{k}(input\%_{i \to k}^{2010})$$ (5)

Where $\sum\nolimits_{k}(output\%_{i \to k}^{2010})$ denotes the correlation coefficient between industry i and its downstream industry k, reflecting the share of industry i's use of inputs to industry k per unit of output, and $\sum\nolimits_{k}(input\%_{i \to k}^{2010})$ is the correlation coefficient between industry i and its upstream industry k, reflecting the share of intermediate products of industry k per unit of industry i's output.

Columns (3) and (5) of Table 9 control for industrial technological advances in this industry. The results show that the coefficients of industrial technological advances on the labour demand of companies in both upstream and downstream industries are significantly negative, indicating the existence of both upstream and downstream transmission effects.

## 6.2 Inhibitory effects: A test of resource taking effect and hiring delay effect

**6.2.1 Resource curse effect.** The previous theoretical analysis that industrial technology advancement produces investment expansion effect and inhibits company labour employment. Enterprises that take the initiative to upgrade industrial technology are bound to increase equipment investment, talent investment and R & D investment during the transition period, for this reason, this paper adopts the enterprise free cash flow measure of enterprise capital capacity, Table 10, column (1) columns of the regression of industrial technological advances on the capital capacity of the enterprise, the results show that industrial technological advances on the coefficient of free cash flow of the enterprise at the level of 5 per cent significantly negative, which indicates that there is an effect of capital hogging.

**6.2.2 Delayed employment effect.** The previous theoretical analysis suggests that the impact of corporate technological advances on companies may have a certain employment delay effect, the explanatory variables are used in the future period and the next two periods of corporate labour employment, the results of Table 10 Columns (2), Columns (3) show that the tfp coefficients are significantly negative at the level of 1%, which suggests that the industrial technological advances significantly inhibit the employment of company labour in the future period and in the next two periods of time, and that there is a hiring delay effect.

**Table 10. Tests for resource usage effects and hiring delay effects.**

|  | (1) | (2) | (3) |
| --- | --- | --- | --- |
|  | **Free cash flow of the company** | **F.lnNumber** | **F2.lnNumber** |
| *tfp* | -30.0091** | -0.6100*** | -0.5682*** |
|  | (-2.3105) | (-22.4067) | (-19.3182) |
| *_cons* | -329.4713 | -7.0287*** | -7.1059*** |
|  | (-1.4776) | (-14.5766) | (-12.8130) |
| *Control* | Yes | Yes | Yes |
| City | Yes | Yes | Yes |
| Year | Yes | Yes | Yes |
| N | 40571 | 35958 | 31689 |
| Adj R-sq | 0.0126 | 0.5917 | 0.5670 |

## 7. Conclusions and insights

At the theoretical level, this paper delves into the channels of action through which technological advance in industry affects labour employment in companies. On the one hand, the inhibitory effect of technological advance in industry on the demand for labour, where jobs in which technology has a greater comparative advantage over labour are replaced, reducing the demand for labour by companies. On the other hand, the incentive effect of technological advance in industry on the demand for labour, the effect may come from both the promotion of productivity of companies and the improvement of production and operation costs of companies by technological advance in industry, so that companies expand the scale of production and thus increase the demand for labour (productivity effect), but also technological advances in industry create jobs in which the labour force is more comparatively advantageous relative to technology (creation effect). The combined impact of technological advances in industry on labour demand therefore depends on a comparison of the trade-offs between disincentive and incentive effects. Based on the theoretical analysis, this paper empirically examines the impact of technological advance in industry on company labour employment and its mechanism of action by using China's A-share listed companies as research samples from 2011 to 2020, and finds that: technological advance in industry significantly reduces company labour employment, and overall technological advance in industry has a suppressive effect on labour employment, the inhibitory effect of technological advance in industry on labour demand is stronger than the incentive effect. At the same time, it shows heterogeneous performance among different labour groups, mainly exerting substitution effect by reducing labour employment for low education level and production personnel; the sub-sample test shows that the negative impact of technological advance in industry on labour employment is more significant in companies with high industry concentration, labour-intensive and capital-intensive companies, high degree of marketization, and those facing high financing constraints. In addition, technological advance in industry raises the overall level of employee wages through the skill premium. The policy implications of the above findings are:

1. China needs to further improve the multi-level social security system and accelerate the reform of the unemployment insurance system in order to resolve the risks posed by technological advance in industry to the labour market. At present, China's unemployment insurance system has the structural contradiction of a large gap in the coverage of unemployment insurance for different groups and a mismatch between the groups covered by unemployment insurance and high unemployment risk groups [52]. With the technological advance in industry, the high unemployment risk groups with low education and low skills are bound to face a greater impact, so it is necessary to improve the coverage and effectiveness of unemployment insurance, and to strengthen the social security of informal work, so that it can play a positive role in realising higher-quality and fuller employment. Rather than increasing the cost of dismissal for companies, reference can be made to the Nordic countries' practice of providing short-term unemployment insurance and retraining opportunities for the unemployed, allowing society to share some of the risk of unemployment among workers while maintaining the mobility and vitality of the labour market.

2. Further improve the relevant employment and training systems and re-employment policies, improve the adaptability of workers with different skills to new technologies, guide the structure of the labour market to match the level of technological development, and transform the labour force's quantitative dividend into a human capital dividend. Improve population mobility policies, reduce institutional barriers to the regional mobility of labour,

further optimize the talent training system, and strengthen the training of "complementary" talents, so as to provide strong human resources support for China's transformation from a large manufacturing country to a manufacturing powerhouse. Promote the establishment of a skills upgrading system that involves collaboration between schools and companies, government and schools, and government and companies, with institutions of higher learning adjusting their relevant specialties to meet market labour demand, while at the same time strengthening general education and increasing the flexibility of highly skilled workers to transform their skills.

3. Promoting a fairer and more competitive market environment and easing the financing constraints of companies will help create more employment opportunities and jobs and achieve high-quality development of the Chinese economy. Relevant departments should appropriately weaken the market control power of large-scale companies in high-monopoly industries, actively guide leading companies in various industries to strengthen vocational training for their employees while applying new technologies, promote effective competition in the market in terms of technology, and alleviate the problem of "technological unemployment" from within the companies, so as to alleviate the impact of robot application on employment under the prerequisite of safeguarding production efficiency.

4. Rationally guiding the conduction effect of technological advance in industry on labour demand within industry and the spillover effect between industries, actively promoting the deep integration of traditional manufacturing, service and other industries with technology, and increasing the training of professionals in industries with low technological complexity, so as to bring the rate of substitution of low-skilled personnel by technology and the rate of filling in of high-skilled personnel into a dynamic equilibrium, and to achieve coordinated development of industries. Policymakers need to pay more attention to the industrial technology-related policies of companies upstream and downstream of the industrial chain to promote mutual benefits and win-win situations. The Government can establish various modes of cooperation, such as setting up joint research centres and promoting cooperation among industries, universities and research institutes, to promote knowledge exchange and technological innovation among different industries.

5. Technological advance in industry has a certain function of regulating income distribution, so the relevant departments can enhance their influence on companies' decision-making on technological application by means of taxation and subsidies, so as to guide companies to pay more attention to the interests of employees in profit distribution.

## Supporting information

**S1 Data.**
(XLS)

## Author Contributions

**Conceptualization:** Jianmin Liu.

**Data curation:** Yifeng Shen.

**Investigation:** Wenye Fan.

**Methodology:** Xiya Wu.

**Software:** Xiya Wu.

**Visualization:** Wenye Fan.

**Writing – original draft:** Jianmin Liu.

**Writing – review & editing:** Yifeng Shen.

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
