## [Decision Letter · Decision Letter 0]

2 May 2024

PONE-D-24-02390Technological Advance in Industry and the Demand for Labour Employment in CompaniesPLOS ONE

Dear Dr. Liu,

Thank you for submitting your manuscript to PLOS ONE. After careful consideration, we feel that it has merit but does not fully meet PLOS ONE’s publication criteria as it currently stands. Therefore, we invite you to submit a revised version of the manuscript that addresses the points raised during the review process.

We look forward to receiving your revised manuscript.

Kind regards,

Dan-Cristian Dabija, PhD

Academic Editor

PLOS ONE

Journal Requirements:

Additional Editor Comments:

Dear authors, while some of the reviewers consider that the paper has merits, others consider the opposite. I invite you to answer to all their suggestions, recommendations and concerns and resubmit the paper.

Cristian Dabija

Reviewers' comments:

Reviewer's Responses to Questions

**Comments to the Author**

1. Is the manuscript technically sound, and do the data support the conclusions?

Reviewer #1: Partly

Reviewer #2: No

Reviewer #3: Yes

2. Has the statistical analysis been performed appropriately and rigorously? 

Reviewer #1: Yes

Reviewer #2: No

Reviewer #3: Yes

3. Have the authors made all data underlying the findings in their manuscript fully available?

Reviewer #1: Yes

Reviewer #2: No

Reviewer #3: Yes

4. Is the manuscript presented in an intelligible fashion and written in standard English?

Reviewer #1: Yes

Reviewer #2: No

Reviewer #3: Yes

5. Review Comments to the Author

Reviewer #1: This paper empirically examines the impact of technological advance in industry on the labour employment demand of companies by taking Chinese A-share listed companies as research samples from 2008 to 2022, and draws many enlightening conclusions. There are some concerns as follows.

1. This article needs to further clarify the difference between this article and the existing literature, because there is a lot of literature studying this issue, such as literature studying the impact of artificial intelligence, automation, digital transformation, etc. on labor demand.

2. This article chooses TFP to measure technological progress, which is certainly possible, but it is recommended that the authors further use other standard methods to measure TFP, such as LP and OP methods, etc., so as to ensure the robustness of the results.

3. The benchmark regression in this article has serious endogeneity problems, and it is recommended that the author select instrumental variables for estimation.

4. The author selects a policy as an exogenous shock and constructs a difference-in-difference model, but the author does not explain well why this policy can represent technological progress.

5. This article did not conduct a mechanism test.

6. There are also some writing errors in this article, for example, double is not required in front of difference-in-difference model.

Reviewer #2: Dear Authors,

Unfortunately, your study is unclear, and I do not see publishing worthy novelty in your paper. Firstly, you chose a general approach to examine the overall industry and technological development. Different industry branches have different development trends and different implications for the employment and demand market. You concentrated your study on Chinese industry and chose companies from two cities in China for data collection. In this case, this study is biased and cannot serve any general conclusion. The mathematical expressions for the data evaluation and analysis are vague and poorly structured. With this stated, I do not have any doubts but suggest rejecting your paper from publishing. For the sake of improvements that you might undertake, here are some of my benevolent suggestions and questions:

The title should be more precise and represent the specifics of your study. Your study concentrates on Chinese industry, and hence, this should be in the title as well, e.g., "Technological Advance in Industry and the Demand for Labour Employment in China"

The general development of technology is known as "Industry 4.0" or the fourth industrial revolution. This should be elaborated and analyzed in the literature review. Also, some papers are already dealing with the upcoming Industry 5.0, and I suggest you consider browsing literature on this topic, too.

Throughout the entire paper, you emphasize the general term "industry". Can you prove that technological development in all industries in China is:

Reported?

Has the same effects that you hypothesized in H1a and H1b?

You have a biased population sampling by selecting A-list companies from two rather greatly populated and developed cities. In statistics, this is considered a doubtful specimen and requires a thorough explanation of why this sample is representative.

In equations (1) and (2), index i is not defined. Mathematically, it should be written as lnNumber(t) if it is in function for a certain year.

You refer to tables with data in the text. However, there are no tables in the attachment or the text.

Regards

Reviewer #3: methodologically, the article is correct. The estimation method used and the robustness check tools are correct. And the results obtained prove the reliability of the study. The conclusions drawn from the numerical analysis are also valid.

6. PLOS authors have the option to publish the peer review history of their article (what does this mean?). If published, this will include your full peer review and any attached files.

Reviewer #1: **Yes: **Tenglong Zhong

Reviewer #2: No

Reviewer #3: No

---

## [Author Response · Author response to Decision Letter 0]

27 Jun 2024

Response to academic editor

1.Please ensure that your manuscript meets PLOS ONE's style requirements, including those for file naming.

Response：Following your suggestions, the authors have modified the article formatting according to the PLOS ONE style template, renaming the files according to the file naming requirements.

2.Please include captions for your Supporting Information files at the end of your manuscript, and update any in-text citations to match accordingly.

Response：As per your request, the authors have added the captions of the supporting information files at the end of the manuscript and updated any in-text citations to match accordingly.

Reviewer #1

1. This article needs to further clarify the difference between this article and the existing literature, because there is a lot of literature studying this issue, such as literature studying the impact of artificial intelligence, automation, digital transformation, etc. on labor demand.

Response：Previous literature is mainly about macro-level technological advances as well as micro-level technological advances, while there is an extreme lack of relevant research on the industrial technology level. In practice, industrial technological advances have caused changes in the demand, structure and quality of labour hiring in companies, and some scholars (Dauth et al., 2018) have begun to pay attention to the possible impact of specific technologies such as robotics applications, artificial intelligence, automation, digital transformation and other technologies applied in companies on the labour market in China, but the impact of industrial technologies in a broad sense on the labour hiring decisions of companies and their mechanisms of action has not yet been carried out. This paper provides an in-depth analysis of the evolutionary process of industrial technology, and refines and summarizes the formation, scope and nature of industrial technological advances, based on which it theoretically extrapolates and empirically examines its impact mechanism on company labour hiring, expanding the study of the microeconomic consequences of industrial technology.

Industrial technology suffers from a nonlinear evolutionary process in which technologies evolve from discrete distributions that do not interfere with each other to highly agglomerative network structures that evolve synergistically (Antonio et al., 2010). The fundamental nature of industrial technology is complexity (Hanieh and Arash, 2023), and industrial technology complexity specifically exhibits the characteristics of non-linearity, unpredictability, divisibility and systemicity (Fang et al., 2023). The relevant literature mainly focuses on the fragmentation of ‘theoretical elaboration’, ‘measurement’ (Mewes and Broekel, 2022) and ‘influencing factors’, and has not yet formed a more systematic theoretical study. A more systematic theoretical system has not yet been formed. Previous research on specific technological advances such as artificial intelligence, automation, digital transformation, etc. has not examined the influence mechanism from the perspective of these characteristic facts. In this paper, the analysis of labour employment based on the characteristics of industrial technological advances is obviously more in line with the logic of reality and the paradigm of theoretical deduction, and it is also an enrichment and supplementation of the original industrial technology theory.

2. This article chooses TFP to measure technological progress, which is certainly possible, but it is recommended that the authors further use other standard methods to measure TFP, such as LP and OP methods, etc., so as to ensure the robustness of the results.

Response：Thanking the experts for their valuable comments, the authors use LP and OP methods to calculate the total factor productivity of companies and take the industry average as a proxy variable for industrial technological advancement, and the results, as shown in columns (5) and (6) of Table 5, show that the regression coefficients of LP and OP are significantly negative at the 5 per cent level, a result which further supports the findings of this paper.

Table 5 Key variable substitution

 （5）

lnNumber （6）

lnNumber

LP -0.1123*** 

 (-11.3974) 

OP -0.1962***

 (-18.9475)

_cons -8.8200*** -8.4147***

 (-69.1163) (-66.7219)

Control Yes Yes

Industry Yes Yes

Year Yes Yes

N 39898 39898

p>|t| p<0.001 p<0.001

Adj R-sq 0.6633 0.6652

3.The benchmark regression in this article has serious endogeneity problems, and it is recommended that the author select instrumental variables for estimation.

Response：In view of your constructive comments, the authors chose as an instrumental variable the distance (IV) from each province and district to the nearest port.the distance from each province and region to the port reflects the front-end, middle-end and back-end locality of the technological division of labour in the industry chain with differences in the market environment, and different technological positions will depend on differences in the market environment. Therefore, foreign market proximity forms an important external environment for industrial technological progress, can screen local industrial technology with high correlation with the formation of industrial technological progress, but does not directly affect enterprise labour employment, and is not correlated with enterprise labour employment and the residual term of the model, which can be used as an instrumental variable for industrial technological progress. The instrumental variable Cragg-Donald Wald F-statistic is greater than the 15% level critical value, indicating that there is no problem of weak instrumental variables, and the p-value is significant at the 1% level in the non-identifiability test and the endogeneity test, i.e. the original hypothesis is rejected. Column (3) of Table 6 presents the results of the second stage regression, where the coefficients of the key explanatory variables are all significantly negative at the 1 per cent level. This indicates that the results remain robust after controlling for possible endogeneity issues.

Table 6 robustness tests

 （2）

tfp （3）

lnNumber

 Instrumental variable test

tfp -10.4175***

 (-3.9922)

IV -0.0027*** 

 (-4.3990) 

_cons 0.7323 1.0328***

 (0.2841) (41.1348)

Control Yes Yes

Industry Yes Yes

Year Yes Yes

N 39900 39900

p>|t| p<0.001 p<0.001

Adj R-sq 0.4407 0.4407

4. The author selects a policy as an exogenous shock and constructs a difference-in-difference model, but the author does not explain well why this policy can represent technological progress.

Response：According to your suggestion, the authors have sorted out the relevant studies on the policies of national demonstration zones for the transfer and transformation of scientific and technological achievements, and on this basis, further explained the relationship between the policies of demonstration zones for the transfer and transformation of scientific and technological achievements and technological advances, as follows: each demonstration zone for the transfer and transformation of scientific and technological achievements has introduced a series of reform measures such as the introduction of international technology, capital investment, technology incubation, and technological output, based on the regional endowment of resources and the advantages of the location. Cultivate and grow innovative industrial clusters to promote the synergistic development of regional industries, give full play to the radiation-driven role of the demonstration zones, promote the transformation of scientific and technological achievements and industrial upgrading in the neighbouring regions, and bring about technological advancement of industries in the region.

5. This article did not conduct a mechanism test.

Response：Modified to add the mechanism test.

6.Mechanism test

6.1 Incentive effects: a test of investment expansion effects and chain transmission effects

6.1.1 Technology investment expansion effect

The previous theoretical analysis suggests that industrial technological advances generate investment expansion effects and increase company labour employment. This paper uses the ratio of company R&D investment to total assets to measure the scale of enterprise investment, table 9, column (1) column (1) of the estimation results of industrial technological advances on the scale of enterprise investment, industrial technological advances in the 1 per cent level is significantly positive, technological investment expansion effect exists.

Table 9 Technology investment expansion effects and chain transmission effects

 (1) (2) (3) (4) (5)

 lnincome lnNumber lnNumber lnNumber lnNumber

 Technology investment expansion effect Upstream pass-through effect (industry i is upstream) Downstream effects (industry i is downstream)

tfp 0.0039*** -0.9529*** -0.8041***

 (4.9987) (-23.7675) (-24.5487)

Upstreamct 0.2553*** 0.5697*** 

 (3.1918) (7.1246) 

Dowmstreamct 0.5668*** 0.7941***

 (5.6408) (7.9499)

_cons 0.0493*** -5.6118*** -4.8197*** -8.0124*** -7.2986***

 (3.1417) (-8.0640) (-7.0170) (-14.9749) (-13.7597)

Control Yes Yes Yes Yes Yes

Industry Yes Yes Yes Yes Yes

Year Yes Yes Yes Yes Yes

N 25605 19762 19762 29630 29630

Adj R-sq 0.1877 0.5353 0.5484 0.6070 0.6150

6.1.2 Industry chain transmission effect

The theoretical analysis in the previous section suggests that through industry chain transmission, labour hiring jobs of industrial technological advances can be transmitted to companies upstream and downstream of the industry chain. This paper identifies the relatively important downstream and upstream industries of each industry according to China's input-output table (2010) as follows: for industry i, if more than 1 per cent of the output of industry i is put into another industry k for use, and the proportion of the output of industry k that is put into industry i is not more than 1 per cent, industry k is considered to be a relatively important downstream industry of industry j, and industry j is a relatively important upstream industry of industry k.

 （3）

The tfpct coefficient γown reflects the impact of industrial technological advances on the labour employment of companies in industry i. The Upstreamct coefficient γup measures the impact of industrial technological advances on the labour employment of companies in the downstream industries of industry i, which is known as the ‘upstream transmission effect’, while the Downstreamct coefficient γdown reflects the impact of industrial technological advances in the upstream industries of industry i on the labour demand of companies in industry i, which is known as the ‘downstream transmission effect’. The coefficient γup of Downstreamct reflects the impact of industrial technological advances in the upstream industries of industry i on the labour demand of companies in industry i, which is called the ‘downstream transmission effect’. In this paper, drawing on Acemoglu et al. (2016), Upstreamct and Downstreamct are constructed as follows:

 （4）

 （5）

Where denotes the correlation coefficient between industry i and its downstream industry k, reflecting the share of industry i's use of inputs to industry k per unit of output, and is the correlation coefficient between industry i and its upstream industry k, reflecting the share of intermediate products of industry k per unit of industry i's output.

Columns (3) and (5) of Table 9 control for industrial technological advances in this industry. The results show that the coefficients of industrial technological advances on the labour demand of companies in both upstream and downstream industries are significantly negative, indicating the existence of both upstream and downstream transmission effects.

6.2 Inhibitory Effects: A Test of Resource Taking Effect and Hiring Delay Effect

6.2.1 Resource curse effect

The previous theoretical analysis that industrial technology advancement produces investment expansion effect and inhibits company labour employment. Enterprises that take the initiative to upgrade industrial technology are bound to increase equipment investment, talent investment and R & D investment during the transition period, for this reason, this paper adopts the enterprise free cash flow measure of enterprise capital capacity, Table 10, column (1) columns of the regression of industrial technological advances on the capital capacity of the enterprise, the results show that industrial technological advances on the coefficient of free cash flow of the enterprise at the level of 5 per cent significantly negative, which indicates that there is an effect of capital hogging.

Table10 Tests for resource usage effects and hiring delay effects

 （1）

Free cash flow of the company （2）

F.lnNumber （3）

F2.lnNumber

tfp -30.0091** -0.6100*** -0.5682***

 (-2.3105) (-22.4067) (-19.3182)

_cons -329.4713 -7.0287*** -7.1059***

 (-1.4776) (-14.5766) (-12.8130)

Control Yes Yes Yes

Industry Yes Yes Yes

Year Yes Yes Yes

N 40571 35958 31689

Adj R-sq 0.0126 0.5917 0.5670

6.2.2 Delayed employment effect

The previous theoretical analysis suggests that the impact of corporate technological advances on companies may have a certain employment delay effect, the explanatory variables are used in the future period and the next two periods of corporate labour employment, the results of Table 10 Columns (2), Columns (3) show that the tfp coefficients are significantly negative at the level of 1%, which suggests that the industrial technological advances significantly inhibit the employment of company labour in the future period and in the next two periods of time, and that there is a hiring delay effect.

6. There are also some writing errors in this article, for example, double is not required in front of difference-in-difference model.

Response：Thank you for pointing out the writing errors, which have been corrected by the author.

Reviewer #2

1.The title should be more precise and represent the specifics of your study. Your study concentrates on Chinese industry, and hence, this should be in the title as well, e.g., "Technological Advance in Industry and the Demand for Labour Employment in China"

Response：As you suggested, the title have be changed to ‘Industrial Technological Advance and the Employment Demand for China's Labour Force—Micro Evidence from Labour Hiring in Companies’.

2.The general development of technology is known as "Industry 4.0" or the fourth industrial revolution. This should be elaborated and analyzed in the literature review. Also, some papers are already dealing with the upcoming Industry 5.0, and I suggest you consider browsing literature on this topic, too.

Response：Thanks to your suggestion, this paper further references literature on robotics applications, artificial intelligence, automation, and digital transformation.

3.Throughout the entire paper, you emphasize the general term "industry". Can you prove that technological development in all industries in China is: Reported? Has the same effects that you hypothesized in H1a and H1b?

Response：H1a and H1b are competing hypotheses. In the field of empirical analysis, we often think about the accuracy and completeness of existing hypotheses, and competing hypothesis testing assists us in selecting the most probable explanation by comparing multiple possible explanations and emphasising the relative likelihood between the hypotheses, thus providing new ideas for strengthening causation and refining the analysis of mechanisms. For example, the article CEO Overconfidence and Bonus Target Ratcheting published by The Accounting Review on 17 May 2024 uses competing hypothesis testing with the following specific hypotheses:

H2a: The degree of asymmetric target ratcheting is greater for overconfident CEOs than for non-overconfident CEOs.

H2b: The degree of asymmetric target ratcheting is smaller for overconfident CEOs than for non-overconfident CEOs.

4.You have a biased population sampling by selecting A-list companies from two rather greatly populated and developed cities. In statistics, this is considered a doubtful specimen and requires a thorough explanation of why this sample is representative.

---

## [Decision Letter · Decision Letter 1]

9 Jul 2024

Industrial Technological Advance and the Employment Demand for China's Labour Force—Micro Evidence from Labour Hiring in Companies

PONE-D-24-02390R1

Dear Dr. Liu,

We’re pleased to inform you that your manuscript has been judged scientifically suitable for publication and will be formally accepted for publication once it meets all outstanding technical requirements.

Kind regards,

Adriana AnaMaria Davidescu, Ph.D

Academic Editor

PLOS ONE

Additional Editor Comments (optional):

The paper meets now the standards for publication.

Reviewers' comments:

Reviewer's Responses to Questions

**Comments to the Author**

1. If the authors have adequately addressed your comments raised in a previous round of review and you feel that this manuscript is now acceptable for publication, you may indicate that here to bypass the “Comments to the Author” section, enter your conflict of interest statement in the “Confidential to Editor” section, and submit your "Accept" recommendation.

Reviewer #1: All comments have been addressed

Reviewer #2: All comments have been addressed

2. Is the manuscript technically sound, and do the data support the conclusions?

Reviewer #1: Yes

Reviewer #2: Yes

3. Has the statistical analysis been performed appropriately and rigorously? 

Reviewer #1: Yes

Reviewer #2: Yes

4. Have the authors made all data underlying the findings in their manuscript fully available?

Reviewer #1: Yes

Reviewer #2: Yes

5. Is the manuscript presented in an intelligible fashion and written in standard English?

Reviewer #1: Yes

Reviewer #2: Yes

6. Review Comments to the Author

Reviewer #1: The authors modified the manuscript well, no more comments. After modification ,this manuscript meets the publication requirement.

Reviewer #2: The changes you incorporated in the paper are well-reflected and the paper is ready for its further steps in the publishing process.

7. PLOS authors have the option to publish the peer review history of their article (what does this mean?). If published, this will include your full peer review and any attached files.

Reviewer #1: No

Reviewer #2: No

---

## [Editor Report · Acceptance letter]

7 Nov 2024

PONE-D-24-02390R1 

PLOS ONE

Dear Dr. Shen, 

I'm pleased to inform you that your manuscript has been deemed suitable for publication in PLOS ONE. Congratulations! Your manuscript is now being handed over to our production team.

Kind regards, 

on behalf of

Professor Adriana AnaMaria Davidescu 

Academic Editor

PLOS ONE